# Quantification of Similarity Relationships According to Parameters of Day Surgery System

**DOI:** 10.3390/ijerph16245048

**Published:** 2019-12-11

**Authors:** Beata Gavurova, Viliam Kovac, Jiri Bejtkovsky

**Affiliations:** 1Technical University of Košice, Letná 9, 04001 Košice, Slovak Republic; viliam.kovac@tuke.sk; 2Research and Innovation Centre Bioinformatics, University Science Park Technicom, Technical University of Košice, Němcovej 5, 04001 Košice, Slovak Republic; 3Tomas Bata University in Zlín, Mostní 5139, 76001 Zlín, Czech Republic; bejtkovsky@utb.cz

**Keywords:** day surgery, healthcare, patient, hospitalisation, hospitalisation ratio, region, regional disparity, similarity, Euclidean distance, the Slovak Republic

## Abstract

Performing day surgery should minimise a number of hospitalisation cases, but its use is determined by many factors. It takes advantage of the latest advances in surgical care, enabling better use of highly costly specialised operating room equipment. This analysis of the day surgery system of the Slovak Republic stands on an examination of the five specialised fields—surgery, gynaecology, ophthalmology, otorhinolaryngology, and urology. The explored period covers the years 2009 to 2017. The whole analysis is divided into the two sections—the youth category and for the adult category. For each case, a hospitalisation ratio is computed. A map visualisation supports the analysis outcome. A quantification of the similarity relationships between the regions is done according to a Euclidean distance approach and it is illustrated through the heat map. The centremost region is the Žilina Region with distance at a level of 1.9821, meaning that it performs as the most similar region to a development of a hospitalisation ratio in the whole Slovak Republic regarding all the examined aspects. The findings introduce an important platform for a creation of regional and national health plans in the area of healthcare provision for the population of the country.

## 1. Introduction

Greater use of day surgery can reduce the utilisation of hospital resources with the added benefit for patients—usually they prefer day surgery as it allows them to return home on the same day. The use of day surgery has increased in all the European Union member countries over the past few decades [1]. It has been allowed due to progress in surgical techniques and anaesthesia. Performing day surgery should minimise a number of hospitalisation cases, but its use is determined by many factors. In the long term, there is not enough pressure from the bottom, that is, from the patients themselves, to minimise the number of hospitalisations in the Slovak Republic. Here, it is seen as an important role for general practitioners, who should be more supportive of patients’ decisions in the cases where the patient’s situation is suitable for day surgery. Transformation of the existing inpatient healthcare facilities into day surgery facilities, as well as the construction of new units, is important for further development of day surgery in the country. In this case, the beds for acute cases are released, a number of the necessary medical staff during the night shift is reduced, and the time needed for surgical procedures and their cost is also reduced. Day surgery takes advantage of the latest advances in surgical and anaesthesia care, enabling better use of highly costly specialised operating room equipment and their supplies. This is also confirmed by the study by Ahmed et al., dealing with the infrastructure for provision of day surgery [2].

Day surgery is one form of saving financial resources of health insurance companies, but its implementation is very expensive. In the recent years, the question of the economic efficiency of introduction of day surgery as well as the transformation of hospital departments into day surgery units has also come to the attention of research teams. This is also confirmed by the results of the study by Gabriel et al., who are concerned with an optimisation of the costs of day surgery performance and a process of finding opportunities for better organisation together with economic and personnel utilisation [3]. These authors also provide an interesting impetus for our subsequent research—an examination the risk factors and causalities associated with surgery and mortality.

The conditions for performing day surgery are different for children than for adults, and they are set in a stricter regime. It also affects the type of diagnoses and comorbidities that are numerous for elderly patients. The same affect can be seen also among the youth population [4]. However, the benefits of day surgery in children outweigh the disadvantages for most children and families. Day surgery reduces the risk of nosocomial infections, the risk of anxiety, disturbed sleep, and so forth. For families, there is minimal disruption to working patterns and childcare needs. It also significantly reduces the economic burden on society. Every day surgery procedure not only provides considerable money savings, but also brings efficiency in medical practice in the healthcare system.

Public policies should make greater efforts to achieve higher transparency and effective accountability, especially in public healthcare systems with a wide range of public and private healthcare service providers. According to many clinical experts, the high variability of day surgery is a partial reflection of the experience and specialisation of the surgical team.

The paper combines the theoretical background of a day surgery field from the other studies mentioned in the successive section devoted to the literature review with the data collected in a territory of the Slovak Republic that can be understood as an area, where no such research has yet been done. A main objective is to potentially create a basement for a future research platform in order to get the day surgery system in the Slovak Republic on a more important level in the healthcare system and to get it more efficient. Primarily, it is very required to carry out the elementary analysis of the day surgery system in the country and to point out the substantial characteristics of the system. Therefore, the paper involves the outcome of the research done in the three main angles of view—a territorial regional examination and a chronological development done from a perspective of the individual specialisation fields of the day surgery system.

## 2. Literature Review

Many foreign research studies deal with the issue of day surgery. Medical studies demonstrate many case reports and share experience from surgical practice. The studies on socio-economic issues and the management of the health processes are highly heterogeneous in this research area. This is also related to an unequal development of day surgery in individual countries, which is also influenced by a form of the health system, its financing, government support, and so forth. The studies considerably dominate the assessment of the day surgery processes, preoperative patient selection, basic procedures, principles of efficient performance, limitation of patients, examination of socio-economic aspects, as well as other determinants of further development in this field.

Heikal et al. draw attention in their study to the importance of preoperative assessment in day surgery and the need to evaluate each health case individually, despite the strong criteria to include the diagnosis in the day surgery regime [5]. Day surgery of children has become more widespread, and more complex operations are being performed on children. The authors emphasise that day surgery improves the hospital efficiency processes and it is cost-effective for them. On the other hand, the provision of high-quality day surgery is subject to many parameters and is very costly to implement. It also requires the availability of specialised medical staff and multidisciplinary medical teams. In addition to the process criteria, the patient-related criteria are important, such as pain management, prevention of post-operative nausea, among other factors. The authors state that the important determinants of the development of day surgery possess a continuous progress in surgical techniques and anaesthesia, as well as the positive expectations of patients and the public in the terms of treatment efficiency.

In their theoretical study, Humphreys and Stocker discuss the convenience of the surgical procedures and point to the possibility of the treating other groups of patients in a field of day surgery for the future [6]. The authors call attention to the social and medical criteria that are limited to the development of day surgery. In many countries, social criteria preclude the development of day surgery, resulting in its underdevelopment and deepening the regional disparities in the numbers of surgical performances and also their types. This fact is also related to a low share of day surgery in the adult category.

An issue of selection of the suitable patients and clinical management of day surgery patients is also discussed by Anderson et al. [7]. In addition to the declared general benefits of day surgery for patients and hospitals, the authors emphasise the importance of a presence of a well-trained motivated multidisciplinary team in the healthcare facilities. The authors describe the importance of the social and surgical factors as well as the preoperative processes. These authors also stress that patients should be judged individually. Hypertension and diabetes mellitus does not perform as a limitation for day surgery, but obesity is a major limitation on the other hand. The National Health Service of England recommends up to 75% of all the surgical procedures to be performed by day surgery.

There is also a trend of an increasing the rate of day surgery due to an increasing technical and technological progress, and also an advancement in anaesthesiology and laparoscopic surgery. Lipp and Hernon point to the significant development of day surgery in the United Kingdom of Great Britain and Northern Ireland [8]. Over the past two decades, the rate of day surgery has increased five times. They also cite the technological development and its impact in improving the surgical and anaesthetic techniques.

A thematically similar study is published by Darwin [9]. The subject of his research interest is also how to carry out the correct selection of patients and the evaluation of suitable procedures for day surgery, including the methods and the techniques applied in anaesthesiology and surgery. The author also emphasises the importance of a multidisciplinary team as a significant benefit for the development of day surgery. He states that day surgery should be done in a separate specialised unit with trained day surgery staff, and that it has to be managed by a clinical leader with a particular interest in day surgery. An equally important matter is preoperative preparation, including selection and evaluation of the appropriate patients and the surgical procedures to treat the particular diagnosis. The study details the principles to select the appropriate patients for surgery as well as the contraindications for day surgery.

Day surgery for children is a subject investigated by Navaratnarajah and Thomas [10]. These authors also confirm the general trends—an expanding range of paediatric patients, which day surgery performance is appropriate for, the importance of a preoperative evaluation, and the selection of the patients according to strict criteria, as well as the flexibility to allow assessment of paediatric patients on an individual basis. Social benefits, as well as economic benefits, are significant for the healthcare facilities and for the whole health system.

Kudchadkar et al. examine the negatives associated with hospitalisation of children after major surgery, highlighting the benefits related to a postoperative treatment in the home environment [11]. The research sample consisted of 221 infants and children from 1 day of age up to 17 years of age received by a hospital intensive care unit.

Some authors investigate the differences in an application of day surgery, for instance Llop-Gironés et al. [12]. The research sample consisted of the hospitals in the Kingdom of Spain, and the data came from the period of 1996 to 2009. The authors examine the time trends of the major surgical procedures differentiated according to the ownership of hospitals, as well as the other classification aspects. The study highlights the differences between the public healthcare entities and the privately owned healthcare entities in their management, financing, and provision of healthcare. During this period, the number of the studies evaluating the efficiency, competitiveness, and quality of healthcare has increased. The involvement of the private actors in the provision of healthcare and hospital management is also visible. The results of the study clearly show an increase in the number of the major surgery performances in private hospitals. Although at the beginning of the explored period, the number of the surgical performances completed was higher in the public hospitals, the situation was reversed at the end of the examined period. The authors call for a need to carry out the comparative analysis to assess the performance of the hospitals as an important tool to ensure the most rational and efficient use of the public resources. According to the authors, the Ministry of Health, Consumer Affairs, and Social Welfare of the Kingdom of Spain has an important role in the process of defining the standards of efficiency assessment. The results of this study contradict the claim that the presence of the private sector in the public health system tends to improve the market.

Many studies are devoted to the guidelines on day surgery as well as the selection of the appropriate patients and procedures. Ng and Mercer-Jones cite the guidelines applied in the United Kingdom, highlighting the importance of the technological progress [13]. This outcome results in expanding a portion of the patients range eligible for day surgery and the possibility of performing more complex procedures that will promote faster patient recovery and better health outcomes. The selection of the patients is also very important according to the Association of Anaesthetists of Great Britain and Ireland, which recommends the implementation of anaesthesia in day surgery under the guidance of a consultant. The association presents a number of recommendations for safe and effective day surgery.

Although day surgery processes have developed in many countries, this development has not been even. The disparities between countries, but also within individual countries, are visible. They are apparent between the particular hospitals and even between the surgeons. This consistent fact is a motive for a study by Leroy et al., aiming at an investigation of a variability in the performance of day surgery among the hospitals in the Kingdom of Belgium [14]. A total of 486 medical procedure data are analysed, along with the expert meetings with medical staff and, in particular, anaesthesiologists from 54 hospitals. As the authors state, day surgery is considered to be very cost-efficient because the hospitalisation time is shortened and the night and weekend medical personnel are eliminated. The results of the study confirm a considerable variability in the performance of day surgery among the hospitals examined. It also reveals a space for better application of day surgery, and a means to increase the efficiency of the treatment processes. In the conclusion, the authors call for a need to investigate the factors that cause the current variability in day surgery, as a further expansion of day surgery is not possible without this knowledge. Moreover, feedback from the hospitals and the healthcare providers and a comparison of the rates of day surgery between the hospitals is very important. It is very important to carry out the benchmarking process in this practise, and it helps to allow regular monitoring of the quality indicators and the factors affecting the unplanned hospitalisations, the postoperative complications, and so forth. According to the clinical experts, a high variability in the performance of day surgery is also due to a varying degree of experience of the specialised surgical teams with a treatment of a considerable mixture of the diagnoses caused by the comorbidities, a higher age of the patients, as well as other aspects. Overall, there is a clear tendency for healthcare facilities to specialise in more serious or less serious cases, which also affects the hospital financial processes. A list of the performances suitable for the day surgery regime, which should be regularly updated, is also noted. An inadequate update is a serious obstacle to the development of day surgery processes in the country.

The aforementioned study is followed by a research performed by Lindqvist et al., who examine the differences in the surgical procedures and practice in day surgery in the eight European countries [15]. They carried out a questionnaire survey, with the assumption is that despite the significant development of day surgery in many countries, the lessons from everyday surgical practice are infrequently disseminated and shared. The results of the study point to a high degree of standardisation of the day surgery regime. They also pay an attention to a role of day healthcare and general practitioners who can make a significant contribution to improvement of the quality of day surgery.

It is clear from the presented studies that very little attention is paid to the regional and geographical aspect of the development of day surgery. This is also the motivation of the research focus of this paper, whose aim is to investigate the regional disparities in the application of day surgery in the regions of the Slovak Republic, evaluating its procedural efficiency by examining the hospitalisation rate for paediatric and adult patients. The analysis outcome would be beneficial to health policymakers and in the creation of a regional healthcare plan.

## 3. Data and Methodology

The methodology, which is applied in the analysis, is aimed at achievement of the defined objectives.

### 3.1. Data

The data come from the database of the Ministry of Health of the Slovak Republic (Ministerstvo zdravotníctva Slovenskej republiky).

From a territorial point of view, a level of the region of the administrative territorial division is selected to be investigated. It embodies the third level of the Nomenclature of Territorial Units for Statistics that serves as a basement for common spatial analysis and it was established by Eurostat—the main statistical office of the European Union [16].

The list of the regions of the Slovak Republic with their abbreviations is as follows:−SK010—the Bratislava Region;−SK021—the Trnava Region;−SK022—the Trenčín Region;−SK023—the Nitra Region;−SK031—the Žilina Region;−SK032—the Banská Bystrica Region;−SK041—the Prešov Region;−SK042—the Košice Region.

The regions are sorted according to their code in an ascending way.

The list of the specialised fields of the day surgery performances is as follows:−specialised field 1: surgery, orthopaedics, surgical emergency, plastic surgery;−specialised field 2: gynaecology and obstetrics;−specialised field 3: ophthalmology;−specialised field 4: otorhinolaryngology;−specialised field 5: urology;−specialised field 6: dentistry;−specialised field 7: gastroenterological surgery, gastroenterology.

The specialised field 1 is abbreviated as surgery, the specialised field 2 as gynaecology, and the specialised field 7 as gastroenterology. It is important to note that the sixth specialised field and the seventh specialised field are omitted from the analysis because of the very low figures of the operated patients. The outcome derived from such a sample of the operations could lead to misleading results. Hence, only the first five specialised fields are analysed.

### 3.2. Methodology

Initially, brief descriptive statistics are offered in the analysis to concisely introduce a topic. The whole analysis demonstrates its outcome in the terms of age of the patients because of the specific needs of youth and adult patients. Each region is introduced with its numbers of operated and hospitalised patients according to their age.

The hospitalisation ratio is calculated as division of hospitalised patients and operated patients within day surgery:(1)$$HR_{r;\;y}=\;\frac{HP_{r;\;y}}{OP_{r;\;y}}$$
where the involved variables mean
−r—the particular region;−y—the particular year;−$HR_{r;\;y}—$a hospitalisation ratio of the r-th region in the y-th year;−$HP_{r;\;y}—$the hospitalised patients in the r-th region in the y-th year;−$OP_{r;\;y}—$the operated day surgery patients in the r-th region in the y-th year.

This hospitalisation ratio is computed for the whole explored period too.

An investigation of similarity of the regions is the successive part of the analytical part of the paper. It is executed through calculation of the Euclidean distance.

The standard form of the Euclidean distance is applied:(2)$$D_{r_{1};\;r_{2}}\;=\;\sqrt{{\left(r_{1}{}_{x}-\;r_{2}{}_{x}\right)}^{2}\;+\;{\left(r_{1}{}_{y}-\;r_{2}{}_{y}\right)}^{2}}$$
where the involved variables indicate
−$r_{1}—$the first region;−$r_{2}—$the second region;−$D_{r_{1};\;r_{2}}—$the mutual Euclidean distance of the r_1_ region and the r_2_ region;−$r_{1}{}_{x}—$the *x* coordinate of the r_1_ region;−$r_{2}{}_{x}—$the *x* coordinate of the r_2_ region;−$r_{1}{}_{y}—$the *y* coordinate of the r_1_ region;−$r_{2}{}_{y}—$the *y* coordinate of the r_2_ region.

Each figure in the paper related to the Euclidean distance is rounded to four decimal places if it is necessary.

Mean similarity of the regions is calculated as a mean value of the mutual distances to all the seven remaining regions. Moreover, a mean value related to the whole explored period is computed as a mean value of all the mean values assigned to the particular region in each year.

As stated above, mean similarity performed as mean distance and it is calculated as follows:(3)$$MD_{r;\;y}\;=\;{\left(n-1\right)}^{-1}\;\overset{n-1}{\underset{i=1}{\sum }}D_{r;\;r_{i}}$$
where the involved variables designate
−r—the particular region;−y—the particular year;−$MD_{r;\;y}—$the mutual mean Euclidean distance of the r_1_ region and the r_2_ region;−i—the particular region of a calculated pair;−n—a number of the regions;−$D_{r;\;r_{i}}—$the mutual Euclidean distance of the r region and the r_i_ region.

There is to note that the order of mentioning the analysed regions in the text is established on the basis of their order according to their classification codes.

## 4. Results

The analytical section is divided into the three main sections from all the potential angles of view.

Firstly, general description of a situation in the individual regions of the Slovak Republic is presented through the main dimension that is represented by a hospitalisation ratio computed for each year of the observed period in every one of the regions. Also, the table with a territorial angle of view is placed at the end of this part of the Results section with a hospitalisation ratio calculated for the whole explored period in every one of all the examined particular regions of the Slovak Republic.

Secondly, similarity of the regions is analysed. This investigation joins itself smoothly to the previous part of this section. It offers a view on the disparities of the individual regions based on the hospitalisation ratio values related to the particular specialised fields of day surgery examined for each individual region separately. Mean similarity of every individual region is computed for the whole explored period too and thus, the centremost region is scrutinised through this approach.

Thirdly, a development of a hospitalisation ratio for the both age categories is explored from a regional aspect. Each region is assigned by the particular specialised field of day surgery for the whole explored period.

### 4.1. Situation Description in the Individual Regions

Firstly, the tables demonstrating the numbers of the operated patients according to their age for the individual regions are presented, as well as those of the hospitalised patients.

Secondly, in order to make the text of the analysis more readable, patients under 18 years of age are called youth patients and patients over 18 years of age are called adult patients. This is not biologically correct, as there are several divisions of youth, but for better understanding of the text it is suitable.

The Bratislava Region performed in a considerably strange way as it is visualised in Table 1. Whilst in the first year of the observed period youth patients kept their hospitalisation ratio at a common level, adult patients reached the highest value of the whole explored period at a level of 20.78%. This is more than two times higher than a mean hospitalisation ratio of that year and also of the Bratislava Region. In the following years, the youth hospitalisation ratio fluctuated significantly, whereas the adult hospitalisation ratio kept its ratio at a common level. Nonetheless, there was no record of youth hospitalisation in the year 2013. In the last explored year, the youth hospitalisation ratio reached its maximum level, standing at 26.17%.

The Trnava Region is characterised by the above mentioned average values of an adult hospitalisation ratio and a fluctuating youth hospitalisation ratio as shown in Table 2. The first two observed years are represented by extremely low numbers. This could possibly be caused by an inconsistency in the original database. However, it is an official statistical outcome. The youth hospitalisation ratio reached its maximum in the middle of the explored period. On the other hand, since the year 2014, the adult hospitalisation ratio has been higher in each year than the mean value of the whole Slovak Republic, except for the year 2014.

The most extreme values overall of hospitalisation ratio are found in the Trenčín Region. These values were reached by youth patients in the period from the year 2014 to the end of the explored period as it is demonstrated by Table 3. Their level peaked over a 50% level, which is extremely alarming. The absolute highest level of 56.21% was reached in the year 2014. On the other hand, the previous years show lower values than the mean value for the whole Slovak Republic with an exception of the year 2010. There is a visible correlation between the increase of both hospitalisation ratios since the year 2014. From the year 2012, each year recorded a higher value of adult hospitalisation ratios than a mean value for the whole Slovak Republic.

The Nitra Region also started with low values, although the highest value of an adult hospitalisation ratio was reached in the year 2010 as illustrated by Table 4. Approximately almost all of the values are similar to the mean value for the whole explored period, whilst only the beginning of the period did not follow this trend. An extreme situation occurred in the year 2010, when the youth hospitalisation ratio was very low but the adult hospitalisation ratio reached a very high value. The opposite situation happened in the last observed year, 2017, when the adult hospitalisation ratio stood approximately at its mean value but the youth hospitalisation ratio peaked at its maximum level of 37.50%.

There were very low figures in the first year 2009 for the Žilina Region in Table 5. A successive development of the youth hospitalisation ratio followed its mean value roughly with an exception in the year 2012 in a form of a steep increase up to a value of 18.32%. The adult hospitalisation ratio advanced with fluctuations, rising until the year 2012 up to its maximum value of 14.65%, then fell until the year 2014 and repeated its rise until the year 2016 with a repeated fall for the last explored year of 2017.

The Banská Bystrica Region perform with extreme numbers as demonstrated by Table 6. The youth hospitalisation ratio is the most fluctuating among all the regions. Its beginning is represented by a high ratio with a significant sink in the next year 2010 and again with a significant rise for the successive year 2011 up to its maximum value of 50.28%. A similar situation is found during the following two years, whilst a steep more than sixteen-time decline occurred in the year 2014. Roughly, very similar values were kept for the years 2015 and 2016 and the end of the explored period is represented by a very significant increase. The adult hospitalisation ratio bore the above mean values until the year 2013, with an exception of the first year, 2009. An interesting fact is that there were recurrent considerable increases and decreases. Since the year 2014, its values also stood under its mean value from an angle of view of the particular years for all the regions of the Slovak Republic.

As it is seen from Table 7, the Prešov Region began with low values for both hospitalisation ratios, as it is also the case for the other regions. The youth hospitalisation ratio kept its figures approximately at its mean value. Considerably lower numbers were reached in the years 2010 and 2013, whilst its maximum value was kept in the last observed year 2017 at a level of 26.70%. A partially different situation occurred for the adult hospitalisation ratio—it reached values that were lower in each year than the mean value of the whole country, with an exception for the year 2010 when it reached only a slightly higher value of 13.34% representing a maximum value of the whole observed period.

The development of the values of the Košice Region had no extreme changes as shown by Table 8, unlike what can be seen in the cases of the previously mentioned regions. The youth hospitalisation ratio kept its values under the mean values of the country, with only two exceptions—in the years 2010 and 2014—where the values were only very slightly higher. Another situation was in the case of the adult hospitalisation ratio whose value for each year was higher than the mean value of the country. Hence, its mean value for the whole observed period is the largest for all the regions.

As it is seen from Table 9 presented above, the youth hospitalisation ratio was higher by four times and the adult hospitalisation ratio by five times throughout the whole explored period, although their difference is equal or lower than two percentage points for the increase by five times. Whilst the numbers for the adult hospitalisation ratio have a common tendency, the youth hospitalisation ratio have a more fluctuating development. The beginning of the period is marked by a considerable decrease between the years 2009 and 2010, when it reached its minimum value at a level of 5.17%, and then by the same increase in an absolute manner towards the year 2011. An approximately two-time rise occurred before the last observed year of 2017, when it reached its maximum value at a level of 25.96%. On the other hand, the adult hospitalisation ratio was at its lowest with a value of 8.84% in the year 2011, and it peaked with a value of 13.42% in the year 2016.

From an angle of view of the individual regions, the best position is kept by the Žilina Region that is on the first place in the terms of youth hospitalisation ratio with a value of 7.00%, and on the second place in the terms of adult hospitalisation ratio with a value of 8.87%. Curiously, the Bratislava Region recorded the same numbers for both age categories at a level of 8.61%. At the same time, it held the best position in terms of adult hospitalisation ratio. On the opposite side, the Trenčín Region performs as the worst one, with a youth hospitalisation ratio at a critical value of 39.29%, being the worst position unconditionally, and an adult hospitalisation ratio of 17.05% that is the second worst position. From the perspective of adult hospitalisation ratios, the last position is kept by the Košice Region with a value of 18.26%, although its mean value of youth hospitalisation ratio stands only at 8.72%. All these result are demonstrated by Table 10.

The subsequent map illustrates the situation of hospitalisation, disregarding the age structure of the patients.

The legend of the map visualised as Figure 1 is as follows—the darker the colour, the higher the value of hospitalisation ratio recorded in the particular region.

The best position is kept by the Bratislava Region, which was followed very tightly by the Žilina Region. The third place is occupied by the Prešov Region. The middle of this rank consisted of the Banská Bystrica Region and the Nitra Region. As it can be seen, a majority of the operations is done for adult patients. Therefore, their numbers have a higher influence on the total numbers. This is why, the Trnava Region and the Košice Region stand at the end of the rank along with the Trenčín Region.

### 4.2. Similarity of the Regions

Similarity of the individual regions is demonstrated on the following heat maps shown as Figure 2, Figure 3, Figure 4, Figure 5, Figure 6 and Figure 7. Each heat map is devoted to the particular class of the day surgery performances during the whole explored period, and there is a summarisation heat map at the end. It shows the total values for the whole period. The darker colour of the cell, the more similar the pair of the regions.

As it can be seen in the visualised heat maps, the individual regions behave very differently in the particular classes of the day surgery performances. Although surgery appear considerably variously, gynaecology, urology, dentistry, and gastroenterology are illustrated more uniformly. The classes ophthalmology and otorhinolaryngology bear an indication of the sharp pattern, but they generally behave in a more varied way.

Surgery has the most assorted pattern. This can be attributed to the high numbers of the operated patients who mainly underwent the performances of this class. It cannot be decided which region behaves differently.

The Košice Region plays an important role in gynaecology. This is due to the very high numbers of hospitalised patients. A maximum of youth patients at a level of 457 in this region and also a maximum for the adult patients at a level of 12,853 causes its extreme position. Successively, there are the two visible clusters—the Bratislava Region, the Trnava Region, and the the Trenčín Region create one, whereas the Nitra Region, the Žilina Region, the Banská Bystrica Region, and the Prešov Region form another.

Ophthalmology has one major region and this was the Bratislava Region. This effect is caused by a very large number of the adult patients operated upon. Its assigned number of hospitalised patients is the highest one, but not in an extreme way in terms of operated patients. The remaining regions behave considerably similarly. The Trnava Region have the lowest numbers of hospitalised patients, but this does not affect the final position effectively.

The Prešov Region is the key region for otorhinolaryngology. It is caused by the highest numbers of operated patients for both age categories among all the regions. Although, the highest numbers of hospitalised patients are found in the Trenčín Region, this do not play such a significant role in the terms of the similarities of the particular regions.

There is one extreme position visible for the Košice Region in urology. This region keeps the most extreme numbers in the terms of youth operated patients and adult hospitalised patients overall. This is quite a strange situation, as it has an extremely low hospitalisation ratio in terms of the youth patients, falling to 2.36%. On the other hand, the adult hospitalisation ratio is the highest, peaking at 41.93%. These figures caused the extreme position of the Košice Region. The other regions behave in considerably various ways.

The last figure demonstrates the similarity throughout all the classes of the day surgery performances. It is clearly visible that the Košice Region holds the most extreme position, whereas the most similar regions are the Banská Bystrica Region and the Prešov Region.

Regarding the values visualised in the previous table, the centremost region from an angle of view of its mutual position with each other region can be decided. Also, a mean value for the whole explored period is calculated and thus, overall position in a two-dimensional space can be demonstrated.

The centremost region is the Žilina Region, with a Euclidean distance at a level of 1.9821, and it is the only explored region whose mean similarity stood under a two-point threshold. This is followed by the Prešov Region with a value of 2.0805 and the Nitra Region with a value of 2.0813. The middle is created by the Trnava Region standing at a level of 2.2177, the Banská Bystrica Region at 2.3947, and the Trenčín Region at 2.4340. On the other side, the most dissimilar regions are the Bratislava Region with a value of 2.5296 and the Košice Region with a value of 2.6984. This outcome confirms an assumption that the biggest cities of the Slovak Republic bear a distinctive position in the field of day surgery.

### 4.3. Development of a Hospitalisation Ratio from a Regional Aspect

The following figures visualise the development of the youth day surgery hospitalisation ratio in the individual specialised fields.

The surgery hospitalisation ratio has no common trending tendency during the explored period as illustrated by Figure 8. A maximum level of 55.06% was reached by the Prešov Region in the year 2011. On the other hand, the zero levels were reached more times, namely, by the Bratislava Region in the years 2010, 2013, and 2017, by the Trnava Region in the years 2009 and 2010, by the Žilina Region in the year 2010, by the Banská Bystrica Region in the year 2011 and 2013, and by the Košice Region in the year 2010.

Gynaecology shows several empty points throughout the whole observed period as it is demonstrated by Figure 9. This is caused by the fact that there were no performances in the Bratislava Region in the years 2011, 2012, and 2014, or in the Trnava Region in the years 2009 and 2013. The riskiest situation arose in the Trenčín Region in the year 2015, where it peaked a level of 94.73%. On the contrary, there were many zero values meaning no riskiness—the years 2010 and 2013 in the Bratislava Region, the year 2010 in the Trnava Region, the years 2009 to 2013 in the Trenčín Region, the years 2009 to 2011, 2014, and 2017 in the Nitra Region, the years 2009 to 2011, 2013, 2014, and 2017 in the Žilina Region, the years 2009 to 2014 and 2016 in the Banská Bystrica Region, and the years 2009 to 2012 in the Prešov Region.

There are several cases with no operated patient for ophthalmology as it can be seen on Figure 10—in the Bratislava region in the years 2009 and 2011, in the Trenčín Region in the years 2009 and 2010, in the Nitra Region in the years 2009, 2015, and 2017, and in the Banská Bystrica Region in the year 2013, whereas there was only a sole operated patient in the Trnava Region in the year 2010 throughout the whole explored period. The maxima were recorded at the highest possible level of 100%—in the Trenčín Region in the year 2014 and in the Nitra Region in the year 2016. In contrast, the lowest possible levels with no hospitalised patients were recorded in the Bratislava Region from the year 2012 to the year 2017, in the Trnava Region in the year 2010, in the Trenčín Region from the year 2011 to the year 2013, in the Nitra Region in the years 2010, 2011, and 2013, in the Žilina Region throughout the whole examined period except for the year 2013, in the Banská Bystrica from the year 2010 to the year 2015, excluding the year 2013, in the Prešov Region throughout the whole explored period except for the final year 2017, and finally, in the Košice Region from the year 2009 to the year 2015, excluding the year 2011.

Otorhinolaryngology possessed a relative turbulent trend shown on Figure 11 during the explored period. A maximum of 80.96% was reached by the Banská Bystrica Region in the year 2011. Generally, this region kept the highest riskiness level foremost, followed by the the Trenčín Region at the end of the observed period. The minimum zero levels were recorded several times—in the Bratislava Region in the years 2012 and 2013, in the Trnava Region in the year 2009, in the Nitra Region from the year 2009 to the year 2011 and, successively, in the years 2014 and 2016, in the Žilina Region from the year 2013 to the year 2015, in the Banská Bystrica Region in the years 2010, 2014, and 2015, in the Prešov Region in the years 2009, 2010, and 2013, and in the Košice Region from the year 2011 to the year 2013.

Urology recorded no operation twice in the explored period as it is visualised by Figure 12—in the Bratislava Region and in the Žilina Region in the first year 2009. The highest hospitalisation rate at a full level was achieved in the Banská Bystrica Region in the final year of 2017. Regarding the general tendency, the urology hospitalisation rate rose undoubtedly. The two cases occurred with no computed hospitalisation ratio because no operations were performed in the Bratislava Region and the Žilina region in the first explored year 2009. The full hospitalisation ratio was present in the Nitra Region and in the Banská Bystrica Region in the final year of 2017. All the remaining parts of the explored period are embodied by a zero level in the Bratislava Region. The same situation is valid for the Trenčín Region, the Nitra Region, and the Prešov Region from the year 2009 to the year 2011, the Žilina Region from the year 2010 to the year 2013 and the final year 2017, the Banská Bystrica Region from the year 2009 to the year 2011 and the years 2013, 2015, and 2016, and, finally, the Košice Region from the year 2009 to the year 2011 and the final year 2017.

The successive figures demonstrate the development of the adult day surgery hospitalisation ratio in the particular specialised fields.

As can be seen from Figure 13 above, the highest level of hospitalisation ratio was reached by the Nitra Region throughout the whole observed period. Understandably, this region also recorded a maximum value of 62.02% in the year 2012. On the other hand, zero levels were recorded by the Trnava Region in the years 2009 and 2010.

The specialised field of gynaecology performed more reasonably for the under 18 years of age category as it is illustrated by Figure 14. The riskiest situation occurred in the Nitra Region in the year 2010, where it peaked a level of 70.23%. Absolutely no hospitalisation happened in the Trnava Region in the year 2010 and in the Trenčín Region in the year 2009.

Ophthalmology performances behave very securely as visualised on Figure 15. It is the most stable specialised field among both age categories, especially for the over 18 years of age category. A maximum value at a level of 27.75% was recorded by the Bratislava Region in the first year 2009, but it was only a rare situation together with the end of the explored period in the Košice Region, where it reached a level of 15.45%. All the other values are lower than a five-per-cent threshold.

Otorhinolaryngology has an oscillating tendency throughout the whole observed period considerably as seen on Figure 16. There was a sole case of no operation in otorhinolaryngology in the Trnava Region in the year 2009. The maximum levels peaked at 98.81% in the Bratislava Region in the year 2010 and at 96.61% in the Banská Bystrica Region in the year 2009. On the other hand, the lowest possible level was reached in the Bratislava Region in the years 2009 and 2013, in the Nitra Region in the years 2009, 2011, 2014, and 2016, in the Žilina Region in the years 2009, 2011, and from the year 2013 to the year 2015, in the Banská Bystrica Region in the year 2010 and from the year 2014 to the year 2017, and, finally, in the Prešov Region in the years 2009, 2010, and 2013.

Urology possessed only one case without operation—the Žilina Region in the year 2009. There was a rising tendency of hospitalisation ratios during the examined period towards its end according to Figure 17. Generally, its peaks are found in the Košice Region with the global maximum of 56.42% in the year 2015. All the regions touch the zero level at least once—the Bratislava Region in the years 2010, 2014, and 2016, the Trnava Region in the years 2009 and 2010, the Trenčín Region from the year 2009 to the year 2011, the Nitra Region in the years 2009 and 2012, the Žilina Region in the years 2010 and 2013, the Banská Bystrica Region in the year 2017, the Prešov Region in the years 2010 and 2011, and, finally, the Košice Region in the year 2009.

## 5. Discussion and Conclusions

Day surgery allows surgery performance to be carried out in a way in which a patient can leave the healthcare facility without staying there or in a hospital and thus, there is no need for sick leave of this patient. The most important evolutionary factors of day surgery are medical, psychosocial, and economic aspects. Day surgery can also have a positive impact on the economy. The patient’s admission to the healthcare facility is determined by several factors. First of all, the clinician assesses the level of an anaesthesia risk, then the patient’s home conditions, a geographical availability of the healthcare facility from the patient’s place of residence, and, finally, the anamnestic information related to postoperative complications in the past. The distance between the patient’s home and the nearest healthcare facility should not exceed 10 to 15 kilometres, so the patient can come to this facility in case of potential postoperative complications. Such occasions usually appear on the fourth or the fifth day. Day surgery facilities have to conclude the contracts with the particular healthcare facilities, which are able to accept a patient with the postoperative complications for hospitalisation if necessary.

Many studies also declare the significant regional disparities in the performance of a number of the day surgery procedures for youths and adults. This is also related to an availability of healthcare service in the country and the particular region, as well as the country health policy [17,18,19,20].

A substantial aim of this paper is to examine the regional disparities in an implementation of day surgery in the Slovak Republic, to investigate the regional disparities in application of day surgery in the individual regions of the country, and to evaluate its procedural efficiency through a hospitalisation rate for paediatric patients and for adult patients separately.

To achieve the goal of the research, a Euclidean distance approach is employed and the outcome is visualised through heat maps which represent the output of the carried-out analysis. The data covering the years 2009 to 2017 are analysed. The results demonstrate several interesting facts. The first two years, 2009 and 2010, possessed records with very low numbers of surgical performances and hospitalisations. The extreme changes of the figures are seen in several cases. For youth patients, they occurred between the years 2010 and 2011 in the Nitra Region and the Banská Bystrica Region, between the years 2011 and 2012 in the Trnava Region, between the years 2011, 2012, and 2013 in the Žilina Region, between the years 2012 and 2013 in the Bratislava Region, and between the years 2013 and 2014 in the Trenčín Region. On the other hand, for the adult patients, such situations occurred between the years 2009 and 2010 in the Nitra Region and the Prešov Region, between the years 2009 to 2012 in the Banská Bystrica Region, between the years 2010 and 2011 in the Trnava Region, and between the years 2010 to 2012 in the Košice Region.

From a countrywide perspective in the youth category, maximum was reached in the last explored year 2017 at a level of 25.96%, whereas it is useless to find a maximum value in the adult category because of the several very near values oscillating around a level of 13%. From a regional viewpoint, a maximum of 39.29% peaked in the Trenčín Region for the youth category. This was a significant extreme compared with the other regions. For the adults, a maximum of 18.26% was recorded in the Košice Region, but the differences between the individual regions were much lower than for the youth category.

The most similar situation to the whole Slovak Republic development of a hospitalisation ratio is found in the Žilina Region. For this finding, a similarity through a Euclidean distance approach is applied. The Žilina Region lies at a distance of 1.9821 points. On the other hand, the Košice Region with a distance of 2.6984, the Bratislava Region with 2.5296, and the Banská Bystrica Region with 2.3947 behave as the most dissimilar regions. The whole situation is shown in Table 11. According to the results illustrated by the heat maps, the following conclusions are reached. In the field of surgery, there is a heterogeneous development of a hospitalisation ratio among the regions. The extreme cases were recorded in the particular fields. In gynaecology, the extreme case is found in the Košice Region. In ophthalmology, the Bratislava Region perform differently from the other regions. In otorhinolaryngology, the Prešov Region behave dissimilarly from the other regions. In urology, the Košice Region perform in an unrelated way to the other regions. In addition to this outcome, the maximum figures are also interesting in the recognition of a situation in the day surgery system. For the youth category, the maximum levels in the individual fields are as follows. In surgery, the Prešov Region peaked at 55.06% in the year 2011, in gynaecology, the Trenčín Region at 94.73% in the year 2015, in ophthalmology, the highest possible level of 100% was reached by the Trenčín Region in the year 2014 and the Nitra Region in the year 2016, in otorhinolaryngology, the Banská Bystrica Region stood at a level of 80.96% in the year 2011, and in urology, the highest possible level 100 % was kept by the Banská Bystrica Region in the year 2017. The adult category has a slightly different situation. In surgery, the Nitra Region peaked at 62.02% in the year 2012, in gynaecology, the Nitra Region at 70.23% in the year 2010, in ophthalmology, the Bratislava Region at 27.75% in the year 2009, in otorhinolaryngology the Bratislava Region at 98.81% in the year 2010, and in urology, the Košice Region at 56.42% in the year 2015.

These findings introduce a very important platform for a creation of regional and national health plans, as well as for a creation of the strategic frameworks in an area of healthcare provision for the population of the Slovak Republic. They perform also as a very important source of information for setting up a network of the new independent units in a form of specialised healthcare facilities for day surgery. The outcome of the analyses demonstrates many regional disparities in the figures and the types of the procedures for both age categories. However, a substantial aim of the Slovak Republic healthcare system is not to ensure a uniform implementation of day surgery, but to instead promote its regulation so that it is available in the different regions, reflecting their demographic structure, the demand for healthcare, the health behaviour of the population, the age structure of the population, and the state of the health indicators in the regions. Besides these points mentioned, the living standards of the patient is important, together with an efficiency of the healthcare procedures of the health system of the Slovak Republic from a macroeconomic aspect.

There are several limitations of this study. Firstly, it is an untouched area of the healthcare service provision in the Slovak Republic. Such an analysis comes from a pioneering research project. A main limitation is assigned to this aspect. The data, which were collected by the healthcare facilities, were only at the level of the performed operations. In future, there is a potential to widen the scope of day surgery research to other fields that are observable. The whole investigation utilises the database of the National Health Information Center (Národné centrum zdravotníckych informácií). The collection of the data was under the supervision of this institution. Another limitation was the market size of the healthcare facilities in the Slovak Republic. There are only several main healthcare institutions, along with a network of minor supporting healthcare facilities with a lower level of hospitalisation capacity.

Further development of day surgery in the country will depend on many factors over the next years. Firstly, there is a financial impact that will be determined by the access of health insurance companies to the day surgery system. Also, government support is a very important component in this field. These companies should support the expansion of a list of performances suitable to be carried out within day surgery, whilst a correct setting of the unit price for the performance to at least cover a share of the cost for the hospitalised patient plays a key role. A further advance of surgical methods as well as anaesthesia care and its impact on miniature invasive surgery and postoperative complications are important to analyse in this process too. Another important determinant is a social factor that affects the length of stay in the healthcare facility after the surgical performance, as well as a choice of this performance in the form of day surgery. The satisfaction of the patients within the implementation of day surgery, all the medical staff with the qualifications for day surgery, as well as the possibilities and means of effective communication of the medical staff with patients are clearly essential in this field.

The experts remain sceptical about the further development of day surgery. This is due to the persistent problems regarding the compensation of the health insurance companies for such surgical procedures, as well as the monthly financial limits, which also create waiting lists in day surgery healthcare facilities. It is necessary to set up the healthcare system, in which the simpler performances are assigned lower payments and more complex performances are assigned higher payments; otherwise, there will be no radical changes that will occur in the development of the day surgery system. If the healthcare facilities are fairly paid for the surgical procedures, they may be willing to get rid of simpler procedures and along with this set up day surgery centres.

## Figures and Tables

**Figure 1 ijerph-16-05048-f001:**
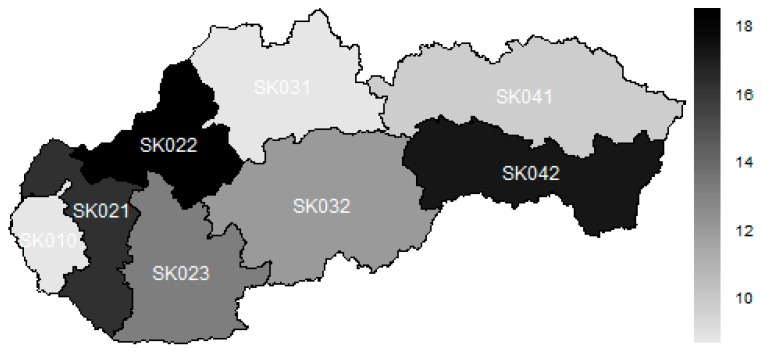
An overall hospitalisation ratio disregarding the age structure for the whole explored period.

**Figure 2 ijerph-16-05048-f002:**
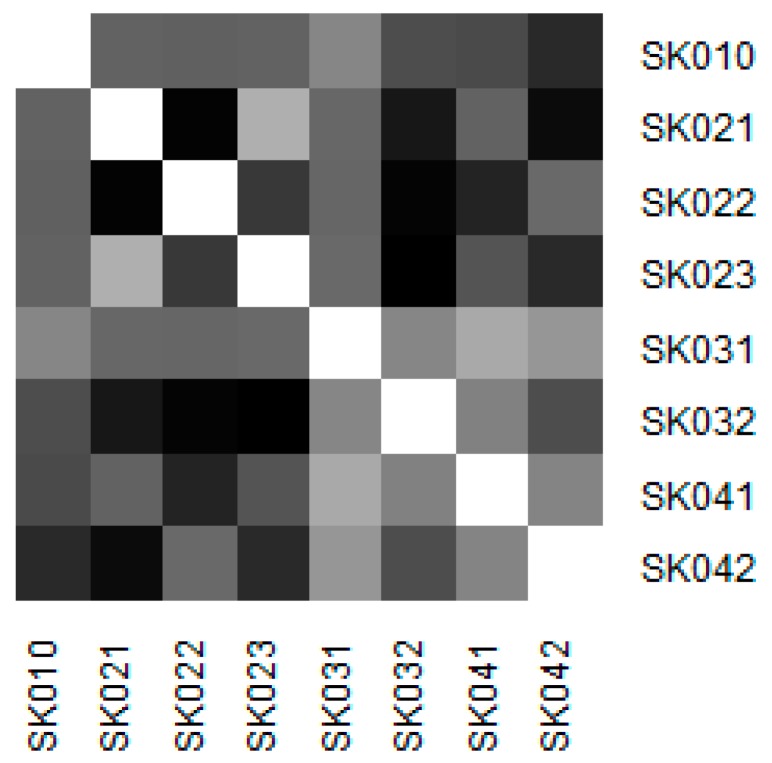
Similarity of the regions in surgery.

**Figure 3 ijerph-16-05048-f003:**
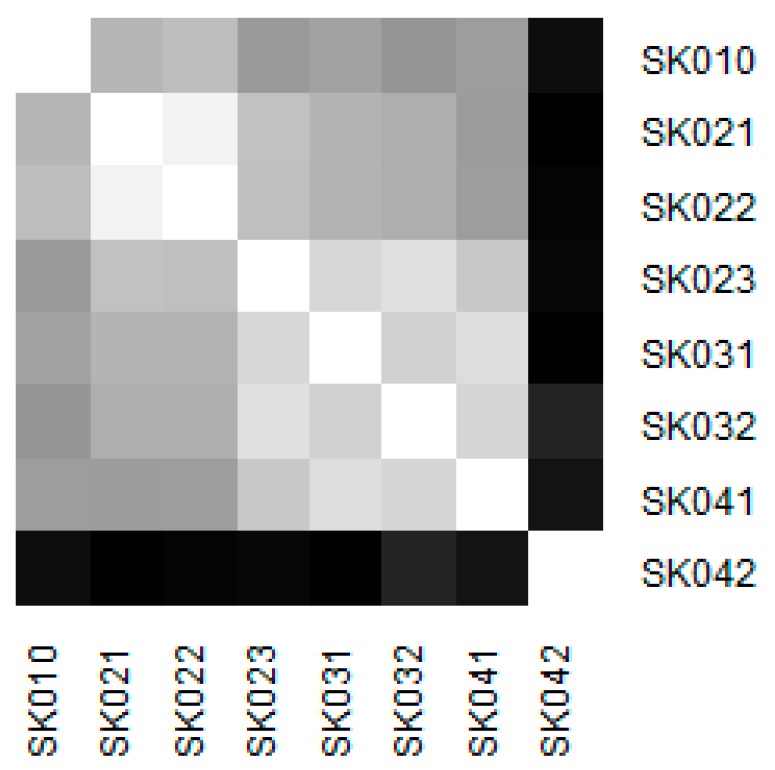
Similarity of the regions in gynaecology.

**Figure 4 ijerph-16-05048-f004:**
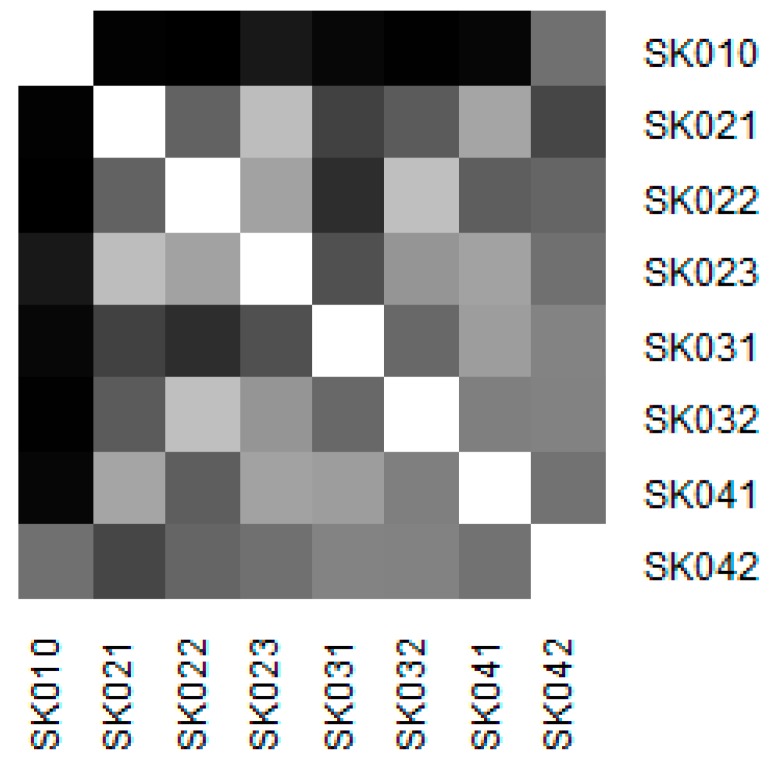
Similarity of the regions in ophthalmology.

**Figure 5 ijerph-16-05048-f005:**
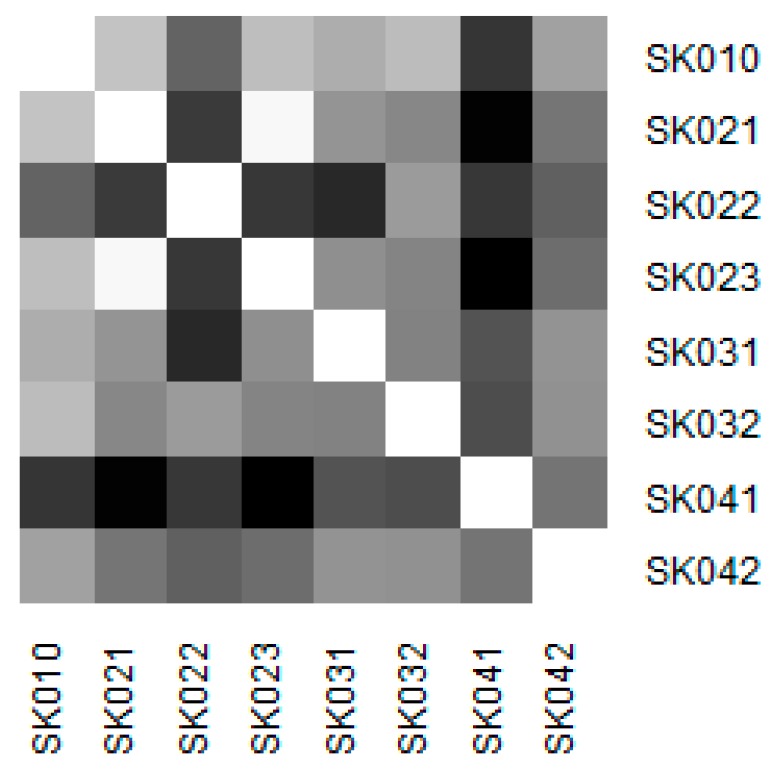
Similarity of the regions in otorhinolaryngology.

**Figure 6 ijerph-16-05048-f006:**
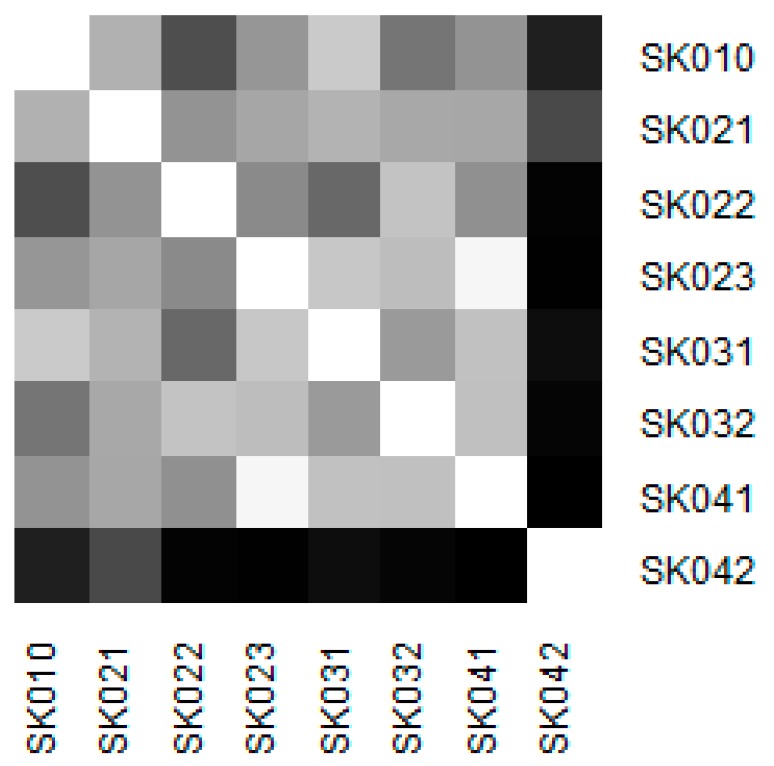
Similarity of the regions in urology.

**Figure 7 ijerph-16-05048-f007:**
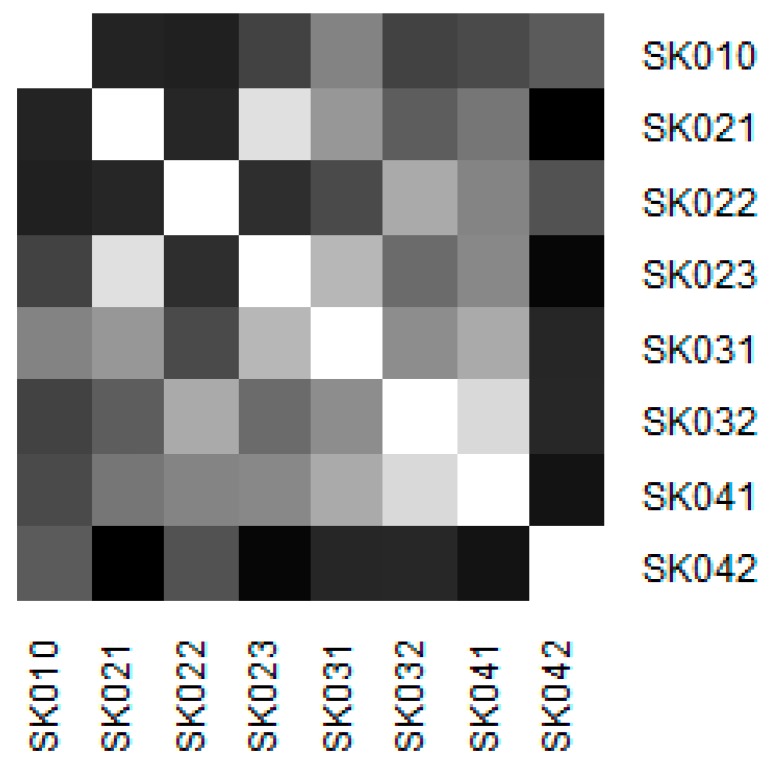
Similarity of the regions in all the classes.

**Figure 8 ijerph-16-05048-f008:**
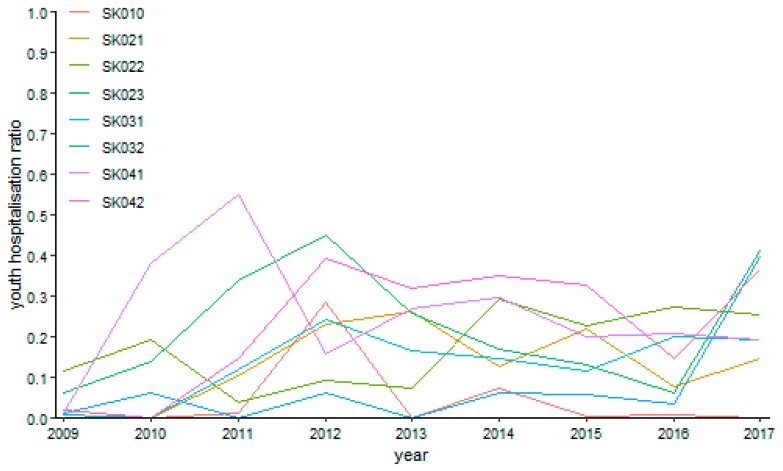
Youth day surgery hospitalisation ratio in surgery.

**Figure 9 ijerph-16-05048-f009:**
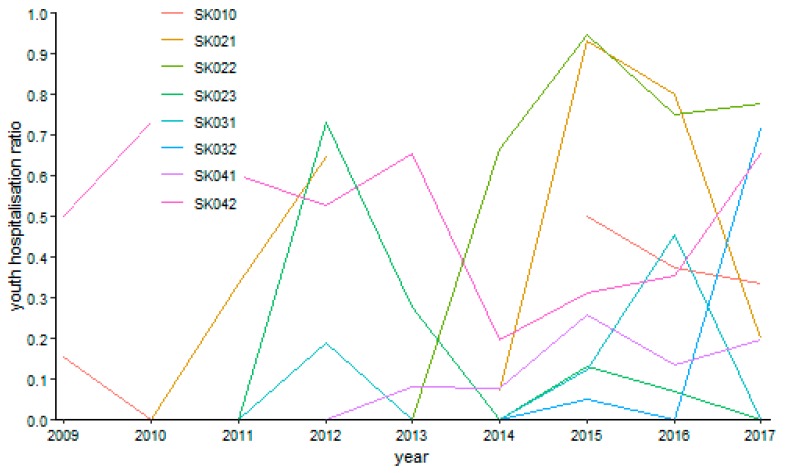
Youth day surgery hospitalisation ratio in gynaecology.

**Figure 10 ijerph-16-05048-f010:**
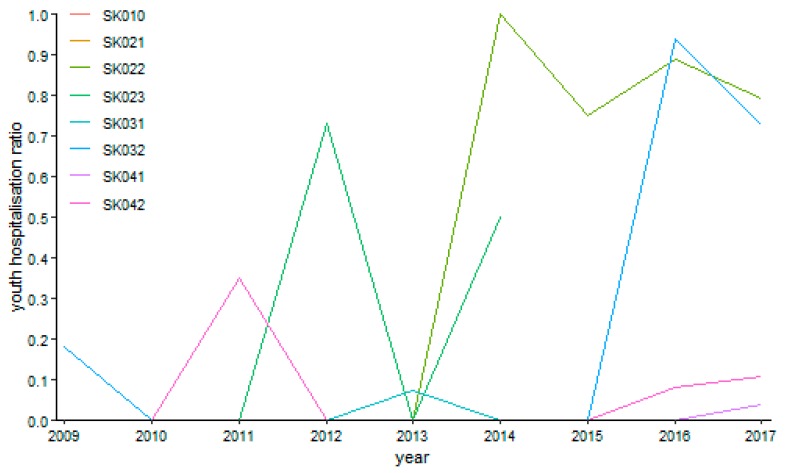
Youth day surgery hospitalisation ratio in ophthalmology.

**Figure 11 ijerph-16-05048-f011:**
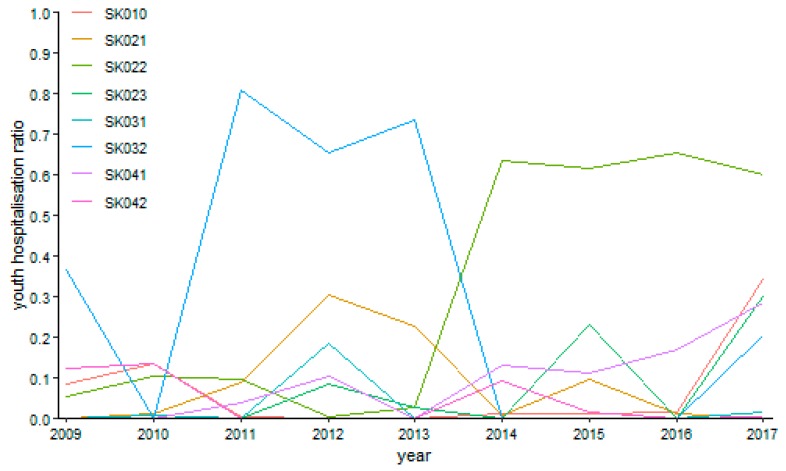
Youth day surgery hospitalisation ratio in otorhinolaryngology.

**Figure 12 ijerph-16-05048-f012:**
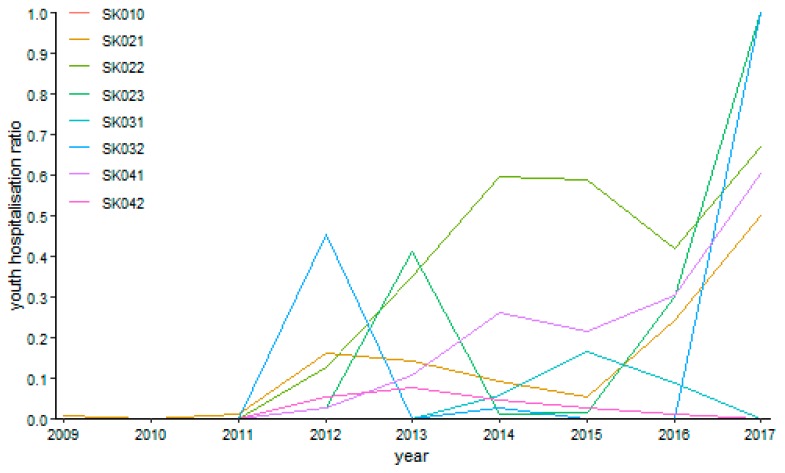
Youth day surgery hospitalisation ratio in urology.

**Figure 13 ijerph-16-05048-f013:**
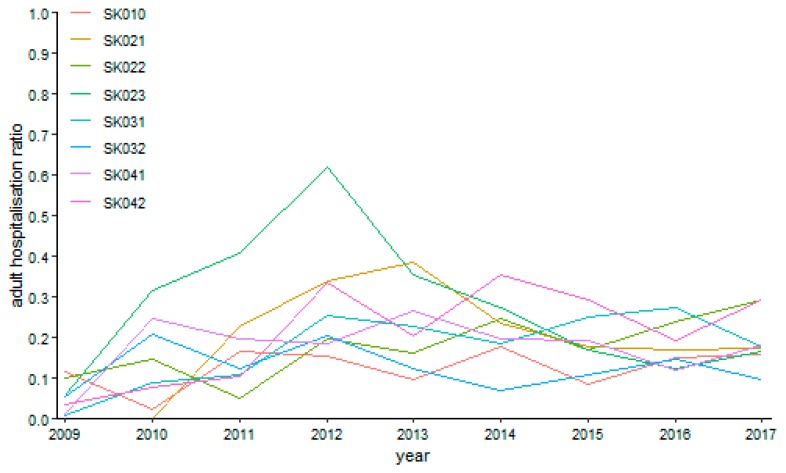
Adult day surgery hospitalisation ratio in surgery.

**Figure 14 ijerph-16-05048-f014:**
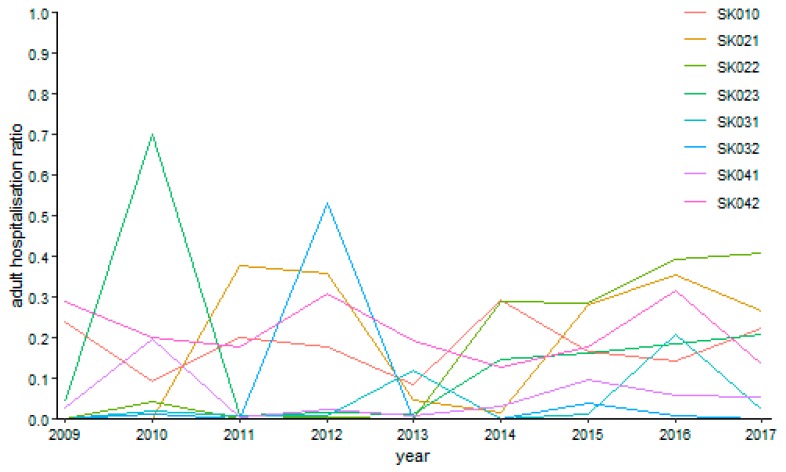
Adult day surgery hospitalisation ratio in gynaecology.

**Figure 15 ijerph-16-05048-f015:**
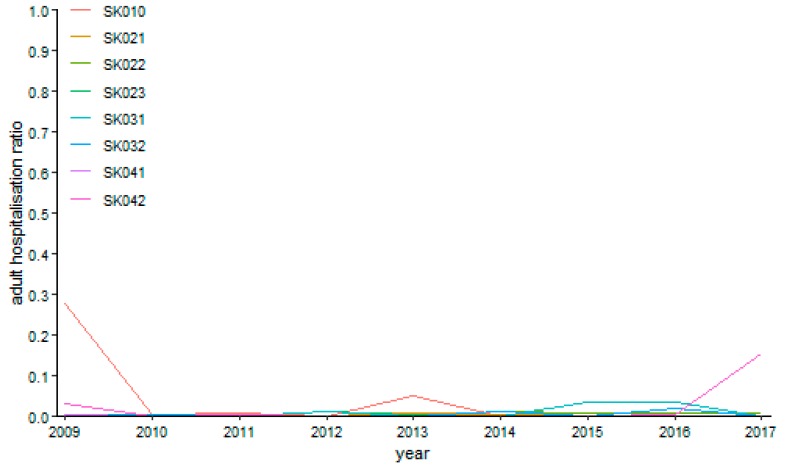
Adult day surgery hospitalisation ratio in ophthalmology.

**Figure 16 ijerph-16-05048-f016:**
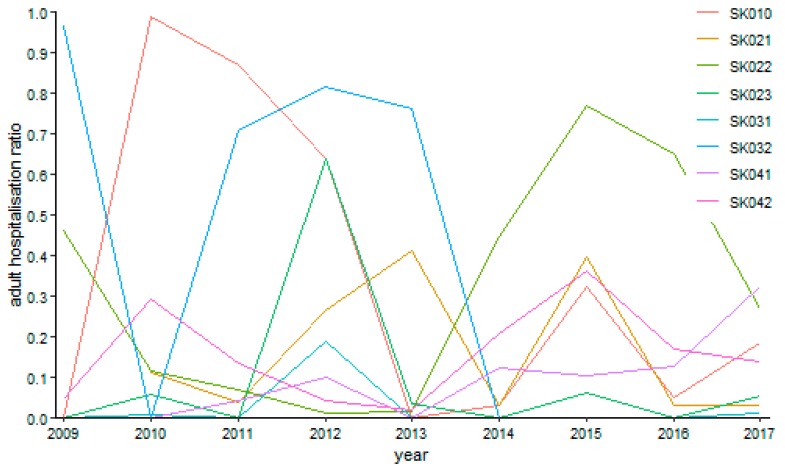
Adult day surgery hospitalisation ratio in otorhinolaryngology.

**Figure 17 ijerph-16-05048-f017:**
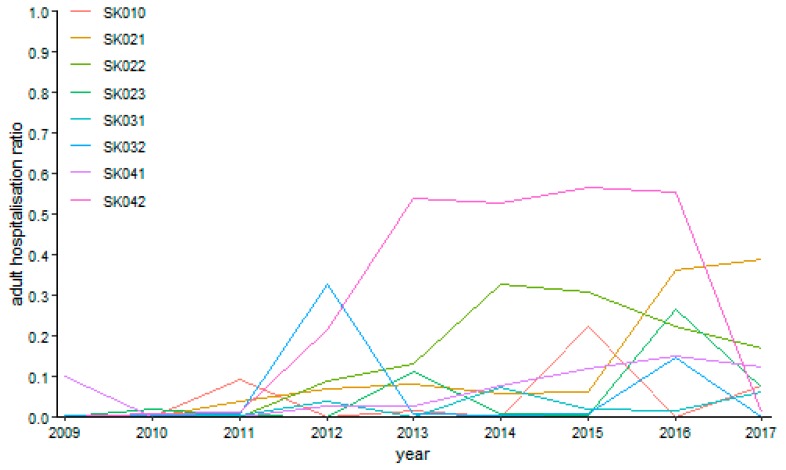
Adult day surgery hospitalisation ratio in urology.

**Table 1 ijerph-16-05048-t001:** Operated and hospitalised patients by age in the Bratislava Region.

Year	Operated Patients		Hospitalised Patients		Hospitalisation Ratio	
	Under 18	Over 18	Under 18	Over 18	Under 18	Over 18
2009	1550	12,932	96	2687	6.19%	20.78%
2010	1096	13,350	124	637	11.31%	4.77%
2011	1152	21,522	6	1965	0.52%	9.13%
2012	1270	25,784	180	1978	14.17%	7.67%
2013	816	25,568	0	1625	0%	6.36%
2014	1295	33,049	26	3118	2.01%	9.43%
2015	1851	35,331	13	2973	0.70%	8.41%
2016	1143	36,747	17	2362	1.49%	6.43%
2017	2358	38,343	617	3535	26.17%	9.22%

**Table 2 ijerph-16-05048-t002:** Operated and hospitalised patients by age in the Trnava Region.

Year	Operated Patients		Hospitalised Patients		Hospitalisation Ratio	
	Under 18	Over 18	Under 18	Over 18	Under 18	Over 18
2009	207	4233	1	2	0.48%	0.05%
2010	313	1705	1	1	0.32%	0.06%
2011	973	8190	88	1795	9.04%	21.92%
2012	932	10,270	247	2861	26.50%	27.86%
2013	497	12,225	103	2323	20.72%	19.00%
2014	1134	14,175	51	1352	4.50%	9.54%
2015	1400	14,803	191	2441	13.64%	16.49%
2016	1482	17,397	98	3229	6.61%	18.56%
2017	1237	19,274	117	3152	9.46%	16.35%

**Table 3 ijerph-16-05048-t003:** Operated and hospitalised patients by age in the Trenčín Region.

Year	Operated Patients		Hospitalised Patients		Hospitalisation Ratio	
	Under 18	Over 18	Under 18	Over 18	Under 18	Over 18
2009	201	4730	14	378	6.97%	7.99%
2010	520	8097	73	861	14.04%	10.63%
2011	843	10,823	61	351	7.24%	3.24%
2012	816	11,753	28	1194	3.43%	10.16%
2013	969	15,526	62	1407	6.40%	9.06%
2014	2028	26,583	1140	5513	56.21%	20.74%
2015	1623	25,879	832	4462	51.26%	17.24%
2016	1913	27,248	1067	6421	55.78%	23.57%
2017	1593	27,204	851	6329	53.42%	23.26%

**Table 4 ijerph-16-05048-t004:** Operated and hospitalised patients by age in the Nitra Region.

Year	Operated Patients		Hospitalised Patients		Hospitalisation Ratio	
	Under 18	Over 18	Under 18	Over 18	Under 18	Over 18
2009	248	6143	1	168	0.40%	2.73%
2010	615	1945	11	571	1.79%	29.36%
2011	514	9253	49	1034	9.53%	11.17%
2012	1218	15,035	201	3777	16.50%	25.12%
2013	734	16,007	89	1994	12.13%	12.46%
2014	884	17,581	39	2361	4.41%	13.43%
2015	977	19,110	162	1950	16.58%	10.20%
2016	765	19,374	78	2077	10.20%	10.72%
2017	800	18,078	300	2096	37.50%	11.59%

**Table 5 ijerph-16-05048-t005:** Operated and hospitalised patients by age in the Žilina Region.

Year	Operated Patients		Hospitalised Patients		Hospitalisation Ratio	
	Under 18	Over 18	Under 18	Over 18	Under 18	Over 18
2009	801	7984	2	20	0.25%	0.25%
2010	1223	3641	5	146	0.41%	4.01%
2011	2142	17,374	83	932	3.87%	5.36%
2012	2898	22,898	531	3355	18.32%	14.65%
2013	2396	18,500	148	2313	6.18%	12.50%
2014	2640	24,131	118	1638	4.47%	6.79%
2015	2774	23,612	152	2023	5.48%	8.57%
2016	2818	24,021	188	2990	6.67%	12.45%
2017	2493	25,415	185	1447	7.42%	5.69%

**Table 6 ijerph-16-05048-t006:** Operated and hospitalised patients by age in the Banská Bystrica Region.

Year	Operated Patients		Hospitalised Patients		Hospitalisation Ratio	
	Under 18	Over 18	Under 18	Over 18	Under 18	Over 18
2009	1706	9380	578	609	33.88%	6.49%
2010	583	3076	13	1063	2.23%	34.56%
2011	1070	13,243	538	1499	50.28%	11.32%
2012	1038	16,950	338	5212	32.56%	30.75%
2013	1138	18,642	472	2403	41.48%	12.89%
2014	3300	22,444	83	1240	2.52%	5.52%
2015	3166	23,626	67	2329	2.12%	9.86%
2016	3101	20,905	88	1494	2.84%	7.15%
2017	3028	27,837	1262	1452	41.68%	5.22%

**Table 7 ijerph-16-05048-t007:** Operated and hospitalised patients by age in the Prešov Region.

Year	Operated Patients		Hospitalised Patients		Hospitalisation Ratio	
	Under 18	Over 18	Under 18	Over 18	Under 18	Over 18
2009	711	6267	1	91	0.14%	1.45%
2010	1357	8575	55	1144	4.05%	13.34%
2011	2817	15,515	303	762	10.76%	4.91%
2012	2520	16,543	269	1258	10.67%	7.60%
2013	3374	19,115	154	1663	4.56%	8.70%
2014	3377	20,966	556	2215	16.46%	10.56%
2015	3372	21,607	453	2616	13.43%	12.11%
2016	3194	24,466	575	1971	18.00%	8.06%
2017	2109	19,318	563	2345	26.70%	12.14%

**Table 8 ijerph-16-05048-t008:** Operated and hospitalised patients by age in the Košice Region.

Year	Operated Patients		Hospitalised Patients		Hospitalisation Ratio	
	Under 18	Over 18	Under 18	Over 18	Under 18	Over 18
2009	1607	14,130	80	1963	4.98%	13.89%
2010	2112	5587	122	1157	5.78%	20.71%
2011	3084	22,398	168	2120	5.45%	9.47%
2012	3054	26,316	324	5258	10.61%	19.98%
2013	2960	28,337	301	4485	10.17%	15.83%
2014	2913	28,040	406	5897	13.94%	21.03%
2015	2749	29,536	240	5959	8.73%	20.18%
2016	2996	29,260	154	6212	5.14%	21.23%
2017	2536	30,277	299	5995	11.79%	19.80%

**Table 9 ijerph-16-05048-t009:** Operated and hospitalised patients by age in the Slovak Republic in the individual years of the whole explored period.

Year	Operated Patients		Hospitalised Patients		Hospitalisation Ratio	
	Under 18	Over 18	Under 18	Over 18	Under 18	Over 18
2009	7031	65,799	773	5918	10.99%	8.99%
2010	7819	45,976	404	5580	5.17%	12.14%
2011	12,595	118,318	1296	10,458	10.29%	8.84%
2012	13,746	145,549	2118	24,893	15.41%	17.10%
2013	12,884	153,920	1329	18,213	10.32%	11.83%
2014	17,571	186,969	2419	23,334	13.77%	12.48%
2015	17,912	193,504	2110	24,753	11.78%	12.79%
2016	17,412	199,418	2265	26,756	13.01%	13.42%
2017	16,154	205,746	4194	26,351	25.96%	12.81%

**Table 10 ijerph-16-05048-t010:** Operated and hospitalised patients by age in the individual regions during the whole explored period.

Region	Operated Patients		Hospitalised Patients		Hospitalisation Ratio	
	Under 18	Over 18	Under 18	Over 18	Under 18	Over 18
SK010	12,531	242,626	1079	20,880	8.61%	8.61%
SK021	8175	102,272	897	17,156	10.97%	16.77%
SK022	10,506	157,843	4128	26,916	39.29%	17.05%
SK023	6755	122,526	930	16,028	13.77%	13.08%
SK031	20,185	167,576	1412	14,864	7.00%	8.87%
SK032	18,130	156,103	3439	17,301	18.97%	11.08%
SK041	22,831	152,372	2929	14,065	12.83%	9.23%
SK042	24,011	213,881	2094	39,046	8.72%	18.26%

**Table 11 ijerph-16-05048-t011:** Mean similarity of the regions throughout the whole explored period.

Year	BL	TA	TC	NI	ZI	BC	PV	KI
2009	2.5205	1.5894	1.4553	1.3263	1.4381	2.7469	1.3387	2.3118
2010	2.7482	2.6357	1.9908	2.0031	2.2450	2.0981	2.1055	2.9046
2011	2.4562	2.2553	2.3802	2.2071	1.9748	2.7012	2.3428	2.8164
2012	2.2880	2.1323	2.7427	1.8641	2.7256	2.2301	2.2011	2.7533
2013	2.4036	2.1465	1.9482	1.8026	1.8028	2.5900	2.2789	3.4554
2014	2.5242	2.5012	2.9656	2.2917	1.8461	2.1106	2.1348	2.4828
2015	2.5706	2.3320	2.8891	2.3489	1.8670	2.0277	2.1991	2.7361
2016	2.7996	2.1246	3.0764	2.3847	1.7750	2.1377	2.1559	2.4545
2017	2.4553	2.2422	2.4579	2.5031	2.1648	2.9097	1.9677	2.3706
whole period	2.5296	2.2177	2.4340	2.0813	1.9821	2.3947	2.0805	2.6984

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
