# Peer review of "Quantification of Similarity Relationships According to Parameters of Day Surgery System"

_ijerph, 2019, doi:10.3390/ijerph16245048_

Round 1
Reviewer 1 Report
The expansion of day case surgery is necessary for social systems all over the world. Besides medical arguments there are mostly economic reasons (development of costs) pushing to a trend towards outpatient treatment.
The above is the focus of the author’s manuscript.
They address the specific situation in the Slovak Republic. Therefore, leading to the first question how and if the results of this study may be applied to the situation in other countries/health care systems.
It remains unclear if the authors consider their article as review or analysis article. In case the analysis article is considered the target neither introduction nor abstract present the hypothesis set up (which should be proven or refuted), which tools were used to perform the analysis and which specific or common conclusions were drawn from their analysis.
The mutual analysis of youth and adult category should be question critically. Especially pediatric operations follow other criteria in the day case surgery setting than day case surgery in adult subjects. Furthermore, this makes the already extensive manuscript even more voluminous.
The introduction focusses on the reasoning of day case surgery which cannot to be determined as focus of the analysis. However, the focus is missing. In the following the authors present a literature review over multiple pages, again the focus on the essence of the authors’ idea is hard to be found, leading a long way around to the actual data.
Data and Methodology: This part is presented comprehensively. Nevertheless, one may already see here that 8 regions with 7 specialized fields plus differentiation between youth and adults create a too wide range for the evaluation volume.
Table one to ten should be compressed to show only the relevant data for the comparison. The same applies for figures 1 to 17.
Compared to the extensive previous pages the discussion is held very short but rather unfocussed.
Especially the conclusion paragraph holds statements about the financial compensation and the experts’ opinion on the development of the day case surgery which were not reviewed in the preceding pages. Therefore, there is a discrepancy to the more descriptive analysis of the current situation.
In general, the subject of the here presented analysis is considered interesting. The presentation is very descriptive and difficult to read. Reasons for the differences between the observed regions are not really investigated. In the here presented form a publication in Environmental Research and Public Health cannot be supported.
Author Response
Dear reviewer
Thank you very much for your valuable comments. Here is a list of the modifications that were done according to your suggestions.
1. Each country has its own specific health system. Hence, it is very difficult to implement solution from one country in another one. A better way is to apply a method used to implement such solution. Moreover, health system with its subset in a form of healthcare system is a very comprehensive system and many legal norms relate to it. Therefore, a question concerning this should follow also such conditions that are raised by legal framework. And yes, you are right, day surgery is meant to save the patient and increase efficiency of treatment. It has a substantial aim to ensure early return of the patient to work and thus, to save health system and social system resources. Standard of living and its level of quality is also a major point associated with faster recovery and return to home environment. An issue rises when potential complementary analyses are suitable to be done – economic data are missing, because the health insurance pricing strategies come into the explored field. The Slovak Republic has the three health insurance companies – one is owned by the state and the remaining two ones are privately owned. The price lists of the provided healthcare services of day surgery varies among these insurance companies, the prices of day surgery operations also depend on the contractual relationships of the health insurance companies with the healthcare providers, and the other parameters. The financial day surgery data are difficult to be accessed and they are considered to be very sensitive for the healthcare providers. Therefore, our ambition is to point out the degree of use of day surgery and the regional disparities to provide the valuable data set in order to create a platform appropriate for policymakers. Successively, they know how our analysis outcome can be applied in order to be completed with the financial data from the healthcare providers about the medical equipment. Also, it is important to ensure that the necessary regulatory and stabilisation mechanisms for the development of day surgery are available and useable they are required to receive the state support.
2. Each study brings the valuable insights into the health system settings. The results of our study provide a relevant picture of the level of regional disparities in a field of day surgery performance that represents a serious obstacle in the development of day surgery and in a process of securing its permanent progression. The analysis outcome can be compared with the economic indicators – for instance, the health system expenditures and their impact on a level of health of the population together with the standard of living parameters, which life expectancy, mortality and morbidity structure, quality of life belong among. This means that our study gives a research potential to determine the international and national benchmarks, and to investigate the connections and the factors that may hinder the development of day surgery in the particular countries. It is also important to eliminate the aspect of the health disparities and not only between the countries, but also within the countries between their regions that is also a priority of the World Health Organization. Therefore, this study gives many incentives for the further potential comparative analysis and to uncover new context and relations of very weak development of day surgery
and the factors affecting this field. Their detection will help to ensure
proportional development of day surgery in the different geographical regions of the countries and thus, to ensure an increasing efficiency and effectiveness of the health system of the individual countries.
The paper is ready to be a research paper in a form of a cornerstone of a potential creation of a platform on a topic of day surgery in the Slovak Republic. Because there is only a small market of the healthcare facilities in the country and there are only the several main healthcare institutions with a network of the minor supporting healthcare facilities, it is very required to begin to explore this field of healthcare provision. This paper combines the theoretical knowledge from the other studies mentioned in the literature review with the data collected in a territory of the Slovak Reublic – an area, where no such research has not yet been done. Yes, you are right. Paediatric operations are based on another criteria. That is why, we divided the whole analysis into the two sections – paediatric and adult patients. Also, we consider our paper as the research paper, because the research methods are applied in order to carry out the analysis – mainly, the statistical methods. The database was obtained under a contractual cooperation with the Ministry of the Slovak Republic, the National Center of Health Information, and the Institute of Health Policy.
3. Yes, paediatric day surgery is a subject to the stricter criteria, but our study is focused to assess a state and a development as well as the regional disparities, not the medical aspects of day surgery. However, thank you for your suggestion, it is our ambition for the future to access this field in our analyses. Nevertheless, it will require access to the deeper structured data, so that we know to evaluate separately development, progression and stagnation of day surgery of the paediatric patients in the different regions, as this aspect will also be involved in the further analyses along with social aspects, mobility in the health system, specialisation of clinics, and so forth.
4. A focus of the paper with the main aim of the analysis was added into the paper at the and of the Introduction section.
5. You are right that it is very comprehensive if 8 regions are analysed in 7 fields for both sexes. On the other hand, separation of whatever angle of view could be a topic of the further paper.
6. There are ten tables, because each one relates to another region and also there is the total table for the whole regions throughout the observed period and also the table for each observed year for the whole country. Originally, we would like to demonstrate only the situation when a hospitalisation ratio is higher than a decided threshold, but we also point to the high fluctuation of its numbers. That is why, we leave the whole tables here. If it is unconditionally needed to lower their number, we will do it. Figures 8 to 17 are different from figures 1 to
7. Which of them are redundant according tou you? Figures from 2 to 7 demonstrate similarity of the individual specialised fields of day surgery, whilst figures 8 to 17 show levels of hospitalisation ratio for these fields.
8. The Discussion section was enriched by the limitations of the study. The conclusions were formulated on the basis of the outputs from the analysis. The recommendations coming from the results of the analyses for health policymakers are clear. Many points and settings affect the pricing strategies of the health insurance companies that are owned by both the private sector and the public sector, so it is very demanding create the particular conclusions and adjustments of the stabilisation and regulatory mechanisms. We have mentioned the process of the stratification of the hospitals in the Slovak Republic, which is in the approval procedure nowadays. The position of the day surgery system may change after this process. Also, an important role in the Slovak Republic health system is played by the DRG system, but it does not enter the day surrgery system and it should be dealt with separately. This is also a point, why the importance of our research studies to address the regional issues and the regional disparities is growing significantly. The outcome will certainly help the regional policymakers in order to construct the health plans in the regional policymaking. The conclusion consists of the considerations and links that are also related to long-term research orientation of the authors of the study in this field. The prices of health insurance companies for operations are subject to change considerably often. Their analysis and linking to performance prices was not the ambition of our study. Moreover, the pricing strategies of the health insurance companies could not be published, so these aspects are mentioned in the discussion section with the potential further research i this field within our research team or as an inspiration for the other international
research teams.
Reviewer 2 Report
Data reported in the paper entitled “Quantification of Similarity Relationships According to Parameters of Day Surgery System” should be of interest; however, the objective(s) of the study is not stringent and clear and should be better stated in the Abstract and in the Introduction section.
The English language requires a revision.
Section “3.2. Methodology”: it is not clear if in the “hospitalisation ratio” the number of operated patients is referred to day surgery or to all surgical procedures performed.
The Results section should be better organized. It is difficult to draw the most important results of the study from the text.
In the Discussion section, strengths and limits of the study should be appropriately addressed.
Author Response
Dear reviewer
Thank you very much for your valuable comments. Here is a list of the modifications that were done according to your suggestions.
1. The main objective of the paper was edited and emphasised in the Abstract and also in the Introduction section as you suggested.
2. There were done several modifications in the English language grammar.
3. The statement about the hospitalisation ratio is modified in order to be clearer for readers. This computation refers to only the day surgery operations. Hence, its name consists of the hospitalisation mention.
4. There are the three main parts in the Results section. The first one is devoted to the description of the situation in the individual regions of the Slovak Republic and the second one deals with the analysis of the similarity of these regions. The third one represents a development of the hospitalsiation ratio from a regional aspect meaning it considers the both age categories from a view of the separate specialised fields of the day surgery in the individual regions of the Slovak Republic. At the beginning of the Results section, the paragraph about the contents of this chapter was added in order to get ready reader how the analysis outcome presentation is built up.
5. The Discussion section was enriched by the limitations of the study. The conclusions were formulated on the basis of the outputs from the analysis. The recommendations coming from the results of the analyses for health policymakers are clear. Many points and settings affect the pricing strategies of the health insurance companies that are owned by both the private sector and the public sector, so it is very demanding create the particular conclusions and adjustments of the stabilisation and regulatory mechanisms. We have mentioned the process of the stratification of the hospitals in the Slovak Republic, which is in the approval procedure nowadays. The position of the day surgery system may change after this process. Also, an important role in the Slovak Republic health system is played by the DRG system, but it does not enter the day surrgery system and it should be dealt with separately. This is also a point, why the importance of our research studies to address the regional issues and the regional disparities is growing significantly. The outcome will certainly help the regional policymakers in order to construct the health plans in the regional policymaking. The conclusion consists of the considerations and links that are also related to long-term research orientation of the authors of the study in this field. The prices of health insurance companies for operations are subject to change considerably often. Their analysis and linking to performance prices was not the ambition of our study. Moreover, the pricing strategies of the health insurance companies could not be published, so these aspects are mentioned in the discussion section with the potential further research i this field within our research team or as an inspiration for the other international
research teams.
Round 2
Reviewer 2 Report
The Authors revised the paper as suggested.